# Laparoscopic Esocardiomyotomy—Risk Factors and Implications of Intraoperative Mucosal Perforation

**DOI:** 10.3390/life13020340

**Published:** 2023-01-27

**Authors:** Abdullah Alkadour, Eugenia Panaitescu, Petre Hoară, Silviu Constantinoiu, Madalina Mitrea-Tocitu, Diana Ciuc, Valeriu-Gabi Dinca, Rodica Bîrla

**Affiliations:** 1General Surgery Department, “Carol Davila” University of Medicine and Pharmacy, 050474 Bucharest, Romania; 2Clinic of General and Esophageal Surgery, “Sf. Maria” Clinical Hospital, 011192 Bucharest, Romania; 3Medical Informatics and Biostatistics Department, “Carol Davila” University of Medicine and Pharmacy, 050474 Bucharest, Romania; 4ENT Department CFR 2 Clinic Hospital, 011464 Bucharest, Romania; 5Faculty of Medicine, “TituMaiorescu” University, 031593 Bucharest, Romania

**Keywords:** laparoscopic esocardiomyotomy, intraoperative mucosal perforation, esophageal leak, intraoperative endoscopy

## Abstract

Background: Mucosal perforation during laparoscopic esocardiomyotomy is quite frequent, and its consequences cannot always be neglected. The purpose of the study is to investigate the risk factors for intraoperative mucosal perforation and its implications on the postoperative outcomes and the functional results three months postoperatively. Material and methods: We retrospectively identified the patients with laparoscopic esocardiomyotomy performed at Sf. Maria Hospital Bucharest, in the period between January 2017–January 2022 and collected the data (preoperative—clinic, manometric and imaging, intra-and postoperative). To identify the risk factors for mucosal perforations, we used logistic regression analysis. Results: We included 60 patients; intraoperative mucosal perforation occurred in 8.33% of patients. The risk factors were: the presence of tertiary contractions (OR = 14.00, 95%CI = [1.23, 158.84], *p* = 0.033206), the number of propagated waves ≤6 (OR = 14.50), 95%CI = [1.18, 153.33], *p* < 0.05), the length of esophageal myotomy (OR = 1.74, 95%CI = [1.04, 2.89] *p* < 0.05), the length of esocardiomyotomy (OR = 1.74, 95%CI = [1.04, 2.89] *p* < 0.05), and a protective factor—the intraoperative upper endoscopy (OR = 0.037, 95%CI = [0.003, 0.382] *p* < 0.05). Conclusions: Identifying risk factors for this adverse intraoperative event may decrease the incidence and make this surgery safer. Although mucosal perforation resulted in prolonged hospital stays, it did not lead to significant differences in functional outcomes.

## 1. Introduction

Achalasia, although the most studied esophageal motility disorder, still has no clear etiology, with numerous hypotheses, ranging from autoimmune to viral infections, and all leading to the destruction of neuronal ganglia in Auerbach’s myenteric plexus. The disease is characterized by the lack of relaxation of the lower esophagal sphincter (LES) during swallowing and the absence of peristalsis of the body of the esophagus [1]. The basal pressure of the lower esophageal sphincter is usually normal or elevated. Clinical presentation includes varying degrees of dysphagia, regurgitation, retrosternal pain and weight loss, all assessed by the Eckardt score (ES). Dysphagia always requires an endoscopic investigation, which, in this case, excluding organic stenosis, raises the suspicion of achalasia. A timed barium esophagogram (TBE), with evaluation at 1 and 5 min, usually shows a dilated esophagus with a “bird’s beak” narrowing at the gastroesophageal junction [2]. The gold standard for the final diagnosis of achalasia is esophageal manometry (ideally high-resolution), allowing classification into three types according to the Chicago Classification [3] with each type having a specific manometric pattern and a slightly different response to treatment [4].

The treatment of the disease includes oral medication (nitrates or calcium channel blockers), with a temporary effect, used only as a bridge to the definitive treatment or for old and frail patients, with reduced life expectancy and a general condition incompatible with a surgical intervention [5]. Upper endoscopy (UE) has always played an important role in the treatment of achalasia, starting with balloon dilation, botulinum toxin injection, passing through the placement of a stent and finally the peroral endoscopic myotomy (POEM) technique, a method that has already proven its effectiveness over time. Surgery has a major, long-standing role in the treatment of achalasia, from double myotomy proposed by Heller [6], via thoracic or abdominal approach, to laparoscopic/robotic anterior esophagogastromyotomy, with anterior/posterior partial fundoplication. All therapeutic methods have variable good results in long term, but some patients may have relapses of the symptoms or develop gastroesophageal reflux.

During any invasive procedure, there are some risks, the most important being esophageal perforation. After endoscopic balloon dilatation, esophageal perforation is a severe complication, with an overall median rate of 1.9% (range 0–16%), in experienced hands (>100 patients treated) [7], solving this complication involving various methods [8,9]

Heller’s esocardiomyotomy implies an extra mucosal divide of the muscular layer on the distal esophagus, the cardia and the proximal stomach, requiring delicate manipulation by preserving only the mucous membrane. The failure of this technique would be related to an insufficient length of the esocardiomyotomy or incomplete sectioning of the circular fibers [10,11].

The incidence of mucosal perforation during laparoscopic Heller myotomy has been reported from 5–33% in cohort studies [12,13,14]. Although this incident is quite frequent, the recognition and intraoperative repair, most of the time, do not have direct consequences on the postoperative evolution of the patient. However, intraoperative mucosal perforation, especially when unrecognized, can lead to a more complicated postoperative course, prolonged hospitalization, increased costs, and decreased quality of life.

The purpose of the study is to identify the risk factors for the occurrence of perforations of the esophageal or gastric mucosa during Heller laparoscopic esocardiomyotomy and to evaluate the implications of this event on the postoperative outcomes and functional results three months after the surgical treatment.

## 2. Materials and Methods

### 2.1. Study Design

Retrospective analysis of a group of patients with achalasia, hospitalized and treated through laparoscopic esocardiomyotomy, associated with an anti-reflux procedure, evaluating the event of intraoperative perforation of the gastric or esophageal mucosa. Different demographic, clinical and paraclinical variables, determined preoperatively, in the patients included in the group, as well as intraoperative data, procedures associated with operative times (intraoperative manometry, intraoperative upper digestive endoscopy, administration of methylene blue on the Fauchet probe) were studied to identify the factors of risk. Postoperative and short-term postoperative follow-up (three months) allowed the collection of data on patients with intraoperative mucosal perforations regarding postoperative complications or functional results.

### 2.2. Patient Recruitment and Data Collection

We retrospectively analyzed the observation sheets and operative protocols of 70 patients with achalasia, who were treated by surgical means, respectively, laparoscopic Heller myotomy and partial fundoplication, in General and Esophageal Surgical Clinic, “Sf. Maria” Hospital, Bucharest, between January 2017 and January 2022. The inclusion criteria were age over 18 years and laparoscopic treatment for achalasia, exclusion criteria were patients who relapsed after laparoscopic, robotic or classical esocardiomyotomy.

The decision of surgical intervention was established in the multidisciplinary team, the young patients (<40 years) being proposed directly to the operation, in the case of the older patients it was their option, taking into account the frequent need for several dilation sessions. Ten patients with relapsed achalasia were excluded, being re-operated by open surgery. Finally, we have included 60 patients. They were evaluated clinically and investigated by endoscopy, timed barium swallow and manometry.

### 2.3. Preoperative Evaluation of the Included Patients

Clinical symptomatology was evaluated with ES. ES has four parameters, dysphagia, regurgitation, retrosternal pain and weight loss, each one ranging from 0 to 3 points, with a maximum of 12 points. A normal Eckardt score is three points or less.

Upper endoscopy (UE) was performed after at least 12 h of fasting, although this interval is not usually enough for esophageal emptying, and esophageal-retained food or stasis is usually encountered. Sometimes, this content is erroneously interpreted as reflux, although a tightly closed cardia is passed with the endoscope. One of the signs that the endoscopist can see is the different color between esophageal stasis (saliva, some foamy liquid) and gastric content (frequently contaminated with yellow bile). The presence of esophagitis or associated gastritis was evaluated.

TBE was performed on the patients included in the study for diagnosis purposes. During the esophageal time, the following aspects were followed: esophageal stasis and the presence of EGJ stenosis, which are the defining elements for achalasia, the presence of tertiary contractions, dilatation of third degree—diameter of the esophagus greater than 6 cm (normal maximum diameter < 2 cm), height of the column of barium in the esophagus >5 cm after 1 min and >2 cm after 5 min [15].

Oesophageal manometry was performed using a pneumohydraulic, low-compliance, water infusion system (Mui Scientific, Mississauga, ON, Canada) with an eight-channel water-perfused catheter (Albyn Medical). The catheter was connected to eight external pressure transducers and the signal was processed using dedicated software (Phoenix V3-Albyn Medical). The following parameters were observed: LES relaxation during swallowing by measuring: LES length, LES basal pressure, LES resting baseline, LES relaxation, relaxation period, LES volume vector and esophageal motility by assessing wave amplitude >35 mmHg, and propagated waves. Based on the manometric criteria, the type of achalasia could be assessed: type I—absent peristalsis without abnormal pressure, type II—absent peristalsis with some normal pressure waves, and type III—absent peristalsis with distal esophageal spastic contractions.

### 2.4. Laparoscopic Esocardiomyotomy

In order to empty the esophagus of solid food, we administrated 500 mL carbonated drink at lunch and strictly liquid diet in the day before the operation. Anticoagulant prophylaxis was performed using low molecular weight heparine in the evening and the naso-esophageal tube was inserted in the morning of the operation.

The surgical treatment was a laparoscopic Heller procedure: Anterior extra mucosal esocardiomyotomy, followed by an antireflux procedure; anterior Dor hemifundoplication; or posterior Toupet fundoplication at 270 degrees. In all the cases, we have used 3D laparoscopy. The esophagus was transhiatal dissected over a length of 8–10 cm, on the anterior aspect. The muscle layer was divided with Metzenbaum scissors, a Hook or with a 5mm vessel sealing device, after the dissection of the muscle fibers from the submucosal layer. The assessment of the length of the myotomy at the level of the esophagus and the cardia was conducted with the intraoperative centimeter. Intraoperative manometry was performed to verify the pressure drop in the LES, secondary to a sufficient esogastric myotomy. Intraoperatively, an upper digestive endoscopy was performed to verify the effectiveness of the esocardiomyotomy and assess the integrity of the esophageal and gastric mucosa, by the detection of a possible esophageal or gastric mucosal perforation.

### 2.5. Management of Patients with Perioperative Mucosal Perforation

The mucosal perforations were observed immediately after the event occurred. When this is not recognized, esophageal or gastric mucosal perforations could be identified either by highlighting air bubbles in the lavage liquid during intraoperative UE or by objectifying the methylene blue solution outside the lumen of the digestive tube [12].

The presence of a millimeter perforation required the suture with separate 4-0 resorbable threads. Checking the tightness of the esophageal or gastric mucosal suture was performed intraoperatively by administering 125 mL of methylene blue–saline solution through the Fauchet probe.

The prevention of gastroesophageal reflux disease, secondary to LES dividing, was carried out by performing an anti-reflux procedure, and in the presence of a suture on the esophageal or gastric mucosa, Dor anterior hemifundoplication was chosen, for covering the suture with the gastric serosa. The intervention ended by inserting a drain tube near the esophagogastric junction (EGJ) and a nasogastric suction tube. The resumption of oral nutrition with liquid food was carried out on the second postoperative day, after a water-soluble contrast control swallow, to rule out digestive fistula. In patients with intraoperative perforation of the mucosa and suture, food was resumed after maintaining the nasogastric tube for an average of seven days, followed by a control swallow. The duration of total hospitalization, the postoperative period and the Intensive Care Unit (ICU) stay were evaluated.

### 2.6. Evaluation of Patients at Three Months Postoperative

The patients included in the batch were either readmitted to the clinic for evaluation or contacted by phone, being questioned about the evolution of the symptoms that make up the Eckardt score, 3 months after the surgical intervention.

The evaluation of the postoperative results was carried out by assessing the degree of symptomatology improvement by calculating the postoperative ES and the difference between the preoperative and postoperative score values. The treatment was considered optimal by obtaining a postoperative ES < 3.

### 2.7. Statistical Methods

For the descriptive statistics, the mean and standard deviation were calculated, respectively, the medians and quartiles for the quantitative variables, and the frequencies and percentages of the qualitative variables. To check the normality, we used the Shapiro–Wilk test, and for the homogeneity of the variants, the Levene test. Preoperative and intraoperative variables associated with mucosal perforation were assessed by Fisher’s Exact Test, Independent Samples T Test, Mann–Whitney U, and Likelihood Ratio, as appropriate. We used the ROC curve to identify threshold values for continuous variables. A logistic regression analysis was performed to identify significant predictors of mucosal perforation during laparoscopic esocardiomyotomy. All analyses were performed using the Using SPSS version 23.0 (IMB Corp, Armonk, NY, USA). All tests were 2-sided, and statistical significance was established at a level of *p* < 0.05.

## 3. Results

### 3.1. Descriptive Analysis of the Group

The demographic characteristics of the patients included in the study group can be seen in Table 1. The group included 60 patients, 24 women and 36 men. The mean age was 50.42 ± 15.67 years with extremes of 20 and 80. A total of 19 patients were smokers (31.7%), and 3 patients had a history of abdominal surgery (5%). Pneumatic esophageal dilatation was performed in the history of 9 patients (15%), on average 1.66 ± 1 sessions/patient, over an average interval of 22.33 ± 22.65 months, and they were sent to surgical treatment due to the persistence or recurrence of dysphagia.

The duration of symptoms averaged 39.03 ± 54.52 months and consisted of dysphagia in 60/60 patients, retrosternal pain in 20/60 patients (33.3%), regurgitation in 42/60 patients (70%), and weight loss in the last six months in 29/60 patients (48.3%).Seven patients presented one symptom, 21 patients had two symptoms, 26 patients had three symptoms, and six patients had four symptoms. Preoperatively ES was on average 6.68 ± 2.13 points. Other symptoms present in four patients were a cough in three patients and nausea in one patient. In the 38 patients who underwent preoperative TBE, the following aspects were encountered during esophageal time: esophageal stasis and presence of EGJ stenosis, in all patients (38/38), esophageal dilatation, in 34/38patients (89.4%), presence of tertiary contractions, in 9/38 patients (23.7%). Grade III esophageal dilatation diagnosed by esophageal barium esophagogram was diagnosed in 4/38 patients (10.5%). The height of the barium column in the esophagus >5 cm after 1 min was found in 37/38 patients (97.4%) and >2 cm after 5 min in 36/38 patients (94.7%).

All patients were examined by UE, with the following aspects being noticed: the presence of stasis in the esophagus in 19/60 patients (31.6%), the presence of esophagitis in five patients and gastritis aspects in nine patients. Preoperative esophageal manometric exploration was performed in 37 patients and the following were monitored: LES length was on average 4.67 ± 1.63, LES basal pressure 20.54 ± 8.24, LES resting baseline 49.47 ± 25.64, LES relaxation on average 68.78 ± 29.86%, relaxation period 1.78 ± 1.85, wave amplitude >35 mmHg had an average of 56.32 ± 26.72, and the LES volume vector averaged 6864.97 ± 5531.10. For the manometric variable the number of propagated waves, we obtained a threshold value ≤6 using the ROC curve with AUC = 0.780, with a specificity of 81.8% and a sensitivity of 75% (Figure 1)

In 48 patients, it was possible to assess the type of achalasia by manometric criteria, in 11 patients the type of achalasia was diagnosed by impedance manometry in other services and by conventional manometry in the rest of the patients—type I was encountered in 16 patients and 32 patients presented type II.

Regarding the preoperative evaluation, 38 patients presented ASA risk 1-2 and 22 patients ASA risk 3-4. All patients underwent laparoscopic extra mucosal esocardiomyotomy: the Heller procedure, followed by Dor anterior fundoplication in 59 patients and Toupet procedure in one patient. The length of the esophageal myotomy averaged 6.25 ± 1.58 cm, the length of the gastric myotomy averaged 2.32 ± 0.68 cm, and the total length of the esocardiomyotomy averaged 8.57 ± 1.52 cm. For the intraoperative variable, the length of the esophageal myotomy had a threshold value of >6 cm using the ROC curve with AUC = 0.769, with a specificity of 65.5% and a sensitivity of 80% (Figure 2).

Another intraoperative variable was the length of esophageal myotomy, and for it we obtained a threshold value >8.5 cm using the ROC curve with AUC = 0.765, with a specificity of 69.1% and a sensitivity of 80% (Figure 3).

Verification of the efficacy of esocardiomyotomy using intraoperative upper endoscopy was performed in 48 patients and intraoperative manometry in two patients. The integrity of the esophageal mucosa, by administration of methylene blue solution, through the Fauchet tube, was performed in seven patients.

Intraoperative, we detected five perforative incidents (8.33%)—four patients with 2–3 mm perforations of the esophageal mucosa and one patient with perforation of the gastric mucosa, which was resolved by suturing with 4-0 resorbable threads. In seven patients, there were other associated surgical procedures: transhiatal diverticulectomy in two patients, cholecystectomy in two patients and umbilical hernia cure in three patients. Postoperative complications were recorded in two patients (3.3%), one developed a digestive fistula on the second postoperative day, which required abdominal reoperation and stenting and the other was complicated with intraabdominal postoperative hemorrhage, with bleeding from the incision for the 5 mm trocar in the right flank. Postoperative mortality was 0%.

Patients had an ICU stay of an average of 1.54 ± 1.37 days, postoperative hospital stay averaged 4.08 ± 2.10 days, and total hospitalization averaged 7.04 ± 3.31 days.

Postoperative ES recorded an average of 1.48 ± 1.28, and the difference of preoperative versus postoperative Eckardt score was an average of 5.20 ± 2.18. In 80.73 ± 23.09% of patients, we recorded symptom improvement at three months postoperatively, and only four patients had an ES > 3 (Table 1).

### 3.2. Statistical Correlations of Esogastric Mucosal Perforations during Laparoscopic Esocardiomyotomy

Analyzing the statistical associations of different demographic criteria, results of imaging or functional explorations, and intraoperative data, in our series, we found that there are no statistical correlations between the occurrence of intraoperative perforations of the esophageal or gastric mucosa during esocardiomyotomy with gender, age, smoking status, history of abdominal surgery, primary pneumatic dilatation, clinical symptoms, pre- or postoperative Eckardt score, presence of esophageal dilatation, endoscopic findings, manometric data, type of achalasia, ASA risk, type of anti-reflux procedure, associated surgery. Intraoperative mucosal perforation was associated with the presence of the tertiary contractions detected by TBE (*p* = 0.03471 (Fisher’s Exact Test)), with the manometric variable—the number of propagated waves ≤6 (*p* = 0.03752 (Fisher’s Exact Test)), with the intraoperative variables—the length of esophageal myotomy (*p* = 0.03961 (Mann–Whitney U test)), the total length of esocardiomyotomy (*p* = 0.04423 (Mann–Whitney U test)), administration of methylene blue (*p* = 0.000343 (Fisher’s Exact Test)), intraoperative upper endoscopy (*p* = 0.00326 (Fisher’s Exact Test)), and with the variables—postoperative days of hospitalization (*p* = 0.00137 (Mann–Whitney U test)) and days of hospitalization (*p* = 0.03437 (Mann–Whitney U test)). (Table 1)

### 3.3. Identification of Risk Factors through Univariate Logistic Regression Analysis

In the univariate logistic regression analysis, we identified the following risk factors for the occurrence of the oesophageal or gastric mucosal perforations during laparoscopic esocardiomyotomy: the presence of tertiary contractions during TBE (OR = 14.0, 95%CI (1.23, 159), *p* = 0. 03321), the number of propagated waves ≤6 (OR = 14.5, 95%CI (1.18, 153), *p* = 0.03579), the length of esophageal myotomy (OR = 1.74, 95%CI (1.04, 2.89), *p* = 0.03196), the total length of myotomy (OR = 1.74, 95%CI (1.04, 2.94), *p* = 0.03478), and a protective factor(OR < 1)—the intraoperative endoscopy (OR = 0.04, 95%CI (0.01,0.38), *p* = 0.00566) (Table 2).

### 3.4. The Implications of Mucosal Perforations in the Evolution of Patients

The implications of the intraoperative perforations in the five patients were as follows: four patients had a simple evolution, and only one patient presented a digestive fistula and developed a subphrenic abscess that required laparoscopic reintervention and the endoscopic insertion of an esophageal stent. The median value of the length of hospitalization for patients without mucosal perforation was six days, and for those with mucosal perforation ten days, the difference having a statistical significance. We also found a significant difference between the duration of postoperative hospitalization in patients with mucosal perforation (median = 8.0 days), compared to those without perforation (median = 3.5 days). More than 50% of those with perforation had over 8.0 days of hospitalization, while 50% of those without perforation had more than 3.5 days. During follow-up, we found that there were no statistically significant differences between the functional results (ES, ΔES) of patients with intraoperative mucosal perforations and those without this event.

## 4. Discussion

The risk factors for the occurrence of perforations of the esophageal or gastric mucosa during laparoscopic esocardiomyotomy, identified by the statistical analysis were: the presence of tertiary contractions in TBE, the number of propagated waves ≤6, length of esophageal myotomy, the total length of the esocardiomyotomy. The presence of tertiary contractions seen in TBE, which are non-propulsive contractions, is common in various esophageal motility disorders not only in achalasia [16]. In patients with achalasia, tertiary contractions can be repetitive, being mainly responsible for retrosternal pain and especially characteristic in vigorous achalasia [17].

Another variable, associated with mucosal perforations, proved to be the number of propagated waves less than six, determined by classical preoperative manometry. It is a functional variable, and not a morphological one, characterizing the motor function of the esophagus, which shows the deep disturbance of the peristaltic function. By determining the threshold value for this variable, we identified the patients with an increased risk of mucosal perforations in the studied group; the patients with profoundly altered motility showed an increased risk of intraoperative mucosal perforation.

Performing intraoperative manometry was not identified to be a risk factor in the occurrence of mucosal perforations, thus proving that this intraoperative procedure, associated with the verification of the efficacy of the esocardiomyotomy, can be performed safely. Moreover, intraoperative UE is confirmed as a method of protection against perforating incidents of the gastric or esophageal mucosa during esocardiomyotomy (OR < 1), the distension of the esophageal mucosa due to endoluminal insufflations allowing better exposure of the mucosa. The administration of methylene blue on the Fauchet probe confirms its role in the detection of hidden perforations and especially in checking the tightness of the mucosal suture.

The implications of mucosal perforations on the subsequent evolution of the patient consisted of longer postoperative and total hospitalization duration, for patients with mucosal perforations, but without determining statistical differences regarding the occurrence of postoperative morbidity, reinterventions or functional results at three months.

The research of the specialized literature reflects a reduced concern about identifying the risk factors of this intraoperative incident, which is nevertheless encountered quite frequently—in our study 8.33%, and other studies even up to 25%. A study from 2009 shows 25% perioperative perforations on 106 patients with laparoscopic esocardiomyotomy, in which only a previous myotomy was identified as a risk factor. Postoperative adhesions and the alteration of the anatomical layers in the esogastric region, secondary to the myotomy, can favor mucosal perforation during the iterative intervention [14]. In our study, patients with a history of esocardiomyotomy and recurrence of the disease were not included.

A study from 2016, on a group of 435 patients with laparoscopic esocardiomyotomy, with 15.4% intraoperative mucosal perforations, identified the following risk factors: Age ≥60 years, disease history ≥10 years, prior history of cardiac diseases, preoperative esophageal transverse diameter ≥ 80 mm, and surgeon’s operative experience with fewer than five cases [18]. In our group, we could not identify any of these risk factors.

The authors of two recent meta-analyses comparing results after laparoscopic and robotic Heller myotomy stated that the only statistically significant difference was intraoperative oesophageal perforation rate [19,20].

Intraoperative recognition and suture of perforation are very important for the occurrence of postoperative complications, such as digestive fistula, which cause prolonged hospitalization, sometimes reinterventions or the need for other endoscopic procedures such as esophageal stents. In our study, 1/5 of the patients with perforations developed a digestive fistula that was managed through endoscopic and surgical means, and we did not record any deaths. Other authors report the appearance of digestive fistulas after suturing the incidental mucosal perforation during esocardiomyotomy—from a series of six patients with mucosal perforations, two patients presented a fistula and one died [21].

In the last decade, ES has been preferred over the Vantrappen classification and the modified achalasia dysphagia score [22]. By current trends, in our study, the clinical symptomatology was assessed by calculating the pre-and postoperative ES.

TBE was proposed to support the diagnosis of achalasia, but also to assess the post-procedural results. The height of the barium column at 1, 2, and 5 min after ingestion of a large bolus of barium is dependent on the retention of barium and the rate of its emptying from the esophagus. Currently, the clinical guideline of the American College of Gastroenterology (ACG) recommends the use of TBE as a complementary method in patients whose manometric results are equivocal or not classic [23]. Another study recently published the recommendations of the European guide, to support the diagnosis of achalasia: barium height of >5 cm at 1 min and >2 cm at 5 min is suggestive of achalasia [24].In our study, we appreciated grade III dilatation in four patients, barium height of >5 cm at 1 min in 37/38 patients barium height of >2 cm at 5 min in 36/38 patients and the presence of tertiary contractions in nine patients—four patients with mucosal perforations.

Although the value of endoscopy in the diagnosis of achalasia is relatively low, exploration is recommended in all patients with the esophageal syndrome, especially to exclude malignant tumors. The endoscopic diagnostic criteria are: the presence of salivary stasis or ingested food, and esophageal dilatation [25], aspects also encountered in our study. Data from the literature suggest that, although esophageal biopsies are recommended in patients undergoing endoscopic evaluation for dysphagia, to assess for eosinophilic esophagitis, they are generally not necessary, if the endoscopic findings are characteristic of achalasia [1]. In our study, we found the endoscopic appearance of esophagitis in five patients, but this aspect was not statistically associated with intraoperative mucosal perforations.

The European guidelines for achalasia recommend esophageal manometry as the golden standard for the diagnosis of achalasia; the manometric characteristic criteria are incomplete relaxation of the LES, in the absence of normal peristalsis [24]. Other authors consider that the diagnosis of achalasia must be confirmed by high-resolution manometry (HRM), which is considered the current gold standard test [26]. HRM has improved spatiotemporal resolution, providing a more intuitive description of contractile and pressure patterns to improve the classification of motor dysfunction originally described using conventional manometry. The subtypes of achalasia are the foundation of the Chicago Classification, and this approach has led to the idea that achalasia is a heterogeneous disease with distinct patterns of pressurization and contraction in the body of the esophagus. Achalasia presents with three distinct manometric subtypes. All three subtypes have impaired LES relaxation, but the distinguishing features are the pattern of pressurization and contraction of the esophageal body. Type I achalasia (20–40% of cases) is characterized by a 100% aperistalsis, with the absence of pan esophageal pressurization at more than 30 mmHg, type II achalasia (50–70% of cases) is characterized by 100% aperistalsis with a pan esophageal pressurization of more than 30 mmHg, and type III achalasia (5% of cases) is characterized by spastic contractions obliterating the lumen with or without periods of pan esophageal pressurization [27].

In our study, based on conventional manometry data, we assessed type I in 16 patients and type II in 32 patients. For conventional manometry, the absence of contraction waves over 30 mmHg is seen in type I achalasia, while the presence of contraction waves over 30 mmHg, is characteristic of type II achalasia. We did not encounter any patients with type III achalasia.

Laparoscopic esocardiomyotomy has become a well-standardized procedure: A five-port laparoscopic approach, mobilization of at least 180° of the anterior side of the gastroesophageal junction and distal esophagus (360° if a Toupet is planned) followed by a full-thickness myotomy that extends to the distal esophagus at least as far as the esophageal hiatus. Some authors recommend avoiding the extension of the myotomy in the free mediastinum, because of the risk of late saccular dilatation of the myotomy site of the esophagus [28].

In our study, the median value of the myotomy length was 6 cm in patients without mucosal perforations and 7 cm in those with mucosal perforations, this variable is a risk factor (OR = 1.74, 95%CI (1.04, 2.89)). Extending the proximal part of the esomyotomy, in the desire to obtain optimal functional results, could favor the perforation at the cranial extremity of the esomyotomy, but, in the patients in the group, the perforation always occurred near the esogastric junction, on the esophageal mucosa at four patients and the gastric mucosa in one patient. The average value of the extra mucosal gastromyotomy in the patients in our study was 2.32 cm and was not proven to be a risk factor in the occurrence of intraoperative mucosal perforations. The traditional 2 cm extension of the myotomy on the anterior stomach wall was challenged by a prospective study by Oelschlager [29], who showed better results with a 3cm extension. The total length of the esocardiomyotomy was on average 8cm in patients without mucosal perforations and 9 cm in those with perforations, this variable is a risk factor (OR = 1.74, 95%CI (1.04, 2.94)). The failure of the classic procedure was linked on the one hand to the insufficient length of the myotomy, and there have been proposals over time to extend this myotomy both on the esophagus and on the stomach, and the other hand, to the incomplete sectioning of the muscle fibers; thus, requiring the verification of the correctness of the myotomy with the help of intraoperative endoscopy or with intraoperative manometry. The current problem of the length of the myotomy is raised by type III achalasia, in which the optimal results are determined by a longer length of this myotomy and, in which, the optimal method of treatment would be the POEM technique that allows a more extensive myotomy than the laparoscopic technique [30].

The occurrence of gastroesophageal reflux disease (GERD) after myotomy is a common problem, and whether an anti-reflux procedure should be performed is a controversial topic, given the risk of postoperative dysphagia after a fundoplication. As early as 2004, the benefit of adding fundoplication was demonstrated in a randomized, double-blind trial comparing myotomy with or without fundoplication [31]. In that study, abnormal acid exposure on pH monitoring was found in 47% of patients without an anti-reflux procedure and in 9% of patients with Dor fundoplication. Heller myotomy with fundoplication was associated with a significant reduction in the risk of gastroesophageal reflux (OR = 0.11; 95%CI (0.02, 0.59)). This study subsequently published 11-year follow-up data on patient-reported symptoms after surgery, with patients reporting similar long-term outcomes in reflux symptom control for both surgeries [32]. Indirect evidence regarding this clinical question comes from a recent meta-analysis published in 2018, which compared POEM and laparoscopic Heller myotomy with fundoplication [33]. The study included 1542 patients who underwent POEM and 2581 patients treated by Heller myotomy with fundoplication. Distal esophageal acid exposure was higher after POEM compared with laparoscopic myotomy with fundoplication (39.0% vs. 16.8%). In all the patients included in the group, after esocardiomyotomy, we performed an anti-reflux procedure—Dor anterior fundoplication in 59/60 patients and Toupet procedure in one patient. In the data from the literature, the type of fundoplication associated with myotomy remains a controversial topic. Dor anterior 180° fundoplication and Toupet posterior 270° fundoplication are most commonly used, but there are rare reports supporting a 360° Nissen fundoplication. Arguments in favor of the Dor procedure include that it requires less hiatal dissection and covers the exposed mucosa of the myotomy site, which may protect in the event of mucosal perforation. Proponents of the Toupet procedure argue that the 270° posterior fundoplication more effectively prevents gastroesophageal reflux and that fixation at the edges of the myotomy helps prevent the myotomy site from healing in the closed position. However, there are no studies yet to provide conclusive support for one or the other procedure, and the choice of fundoplication continues to be a matter of surgeon preference [34].

In the patients included in the study, intraoperative UE was performed to verify the complete sectioning of the circular fibers of the LES and the integrity of the esogastric mucosa after myotomy and conventional manometry to verify the decreased pressure at the LES level after myotomy. The administration of methylene blue, on the Fauchet probe, to detect mucosal perforation or mucosal suture tightness, was another intraoperative control method, similar to the literature data [14]. Other authors did not practice methods of intraoperative control of the integrity of the esogastric mucosa during esocardiomyotomy [18]. Intraoperative, we detected five mucosal perforations, four at the level of the esophageal mucosa and one at the gastric level. The intraoperative identification of these events allowed the immediate solution by suturing the mucosa and covering the suture with gastric serosa of the anterior Dor hemifundoplication. Postoperative complications were presented by two patients—one with a digestive fistula—although the perforation was recognized and sutured immediately, which required reintervention for drainage and the endoscopic insertion of an esophageal stent, and another patient with postoperative hemoperitoneum, from the left hypochondrium trocar incision, which required reintervention for hemostatic purposes. The duration of postoperative hospitalization was longer in patients with mucosal perforations, 8.0 days compared to 3.5 days in those without perforations, explained by the maintenance of the nasogastric tube and the resumption of oral feeding. In patients with perforations and mucosal suture, the probe was maintained for an average of seven days, to prevent the occurrence of digestive fistula, although, in our patient with this complication, it was manifested by the appearance of digestive contents on the drain tube’s second day postoperatively. The presence of mucosal perforations did not lead to statistically significant differences regarding the occurrence of postoperative complications, surgical reinterventions, or hospitalization in the ICU. We also did not find that the functional results at three months (ES and ΔES) are statistically different in patients with mucosal perforations compared to those without perforations, thus making it clear that mucosal injury itself did not affect the postoperative outcomes of patients.

In our study, over 80% of patients declared symptom improvement at three months postoperatively, results comparable with the data from the literature. A meta-analysis that included 39 studies on laparoscopic myotomy, and a total of 3086 patients, reported an improvement in symptoms in an average of 89% of patients (range 77–100%) [35].

ES is a simple measure designed to track outcomes after achalasia intervention and is currently the standard score, used in almost all treatment trials [36]. Most treatment studies show that ES will improve after the intervention, and higher scores after intervention are associated with more preprocedural symptoms. Regarding the clinical results obtained, we found that the ES improved from an average value of 6.68 preoperatively to 1.48 postoperatively, but four patients (6.66%) had suboptimal ES, according to data from the literature, which considers a suboptimal value a postprocedural ES > 3 [37]. In 2018, some authors systematically evaluated ES and concluded that this score presents a marginal level in terms of reliability and validity and that most of the score could be explained only by the dysphagia component. ES performance appears to be reduced by the inclusion of the chest pain and weight loss component. Thus, it is suggested that ES alone is not sufficient to define the success or failure of the treatment, and it is necessary to develop a new tool to assess patient-reported outcomes [38].

TBE is an important tool in assessing post-therapeutic outcomes [39]. Before therapy, most patients had barium retention at 1 and 5 min after ingestion of a large bolus of barium; after a successful intervention, TBE is expected to show complete emptying of the esophagus. Some studies have suggested that the post-interventional barium esophagogram is a useful tool for assessing outcomes and the need for therapy for treatment failure [40,41]; however, other studies argue against this predictive value [42,43]. Postoperatively, high-resolution manometry can evaluate the completeness of the myotomy, and can detect the presence of spastic contractions; however, it is not able to accurately determine bolus retention or the contribution of gastroesophageal reflux in the persistence of symptoms. Data from the literature are limited regarding the utility of this exploration as a predictive tool in the assessment of treatment failure [28]. Persistence of symptoms or recurrence after treatment of achalasia may have various causes: incomplete myotomy, gastroesophageal reflux with/without obstructive sequelae, fundoplasty failure, persistent spasm of the esophageal body and, in addition, a mechanism unique and problematic, pseudodiverticular dilatation at the place of the mediastinal myotomy. A lack of response to treatment may lead to the consideration of an inaccurate initial diagnosis, making it often necessary to review the basic tests with extra care. Thus, some authors typically pursue a comprehensive, stepwise evaluation involving TBE, high-resolution manometry, and sometimes ambulatory esophageal pH testing in an attempt to identify a specific etiology for the symptoms. Finally, when the specific mechanism for symptom recurrence after achalasia therapy is identified, targeted therapy can be pursued [44].

The limitations of the study may be related to the mono-institutional, retrospective nature of a relatively small sample size. However, in this study, all data were collected in a standardized system. It was very difficult to collect enough patients with laparoscopic esocardiomyotomy for achalasia in a single institution because the prevalence of this disease is low. Some patients had incomplete preoperative data or investigations performed in other institutions, which may introduce some bias. However, we were able to collect preoperative data for a sufficient number of patients to fulfill the proposed goal. Therefore, we believe that our results allow us to draw some pertinent conclusions. Another limitation is the relatively short follow-up of the patients to evaluate the results obtained, but we consider that the implications of the mucosal perforations appear immediately postoperatively or may alter the functional results of the esocardiomyotomy, which we usually evaluate first at three months postoperatively. Additionally, the evaluation of the results for the patients in the study was only performed by ES, not by the other paraclinical methods. The results obtained in our study require verification of prospective studies or external validation.

## 5. Conclusions

Intraoperative mucosal perforations during laparoscopic Heller myotomy occur quite frequently, and their immediate detection and resolution are essential.

Identifying risk factors for this adverse intraoperative event may decrease the incidence and make this surgery safer. The risk factors identified by our study were: the presence of tertiary contractions during TBE, the number of propagated waves ≤6, the length of esophageal myotomy, and the total length of myotomy, which leads to the conclusion that mucosal perforation could be prevented by recognizing the high-risk factors by the surgeon and paying more attention to patients with profoundly impaired esophageal motility and on the other hand limiting esophageal myotomy to 6 cm. Intraoperative endoscopy did not prove to be a risk factor for intraoperative mucosal perforation, but a protective one.

Although mucosal perforation resulted in prolonged hospital stays, it did not lead to significant differences in postoperative morbidity or functional outcomes at three months.

## Figures and Tables

**Figure 1 life-13-00340-f001:**
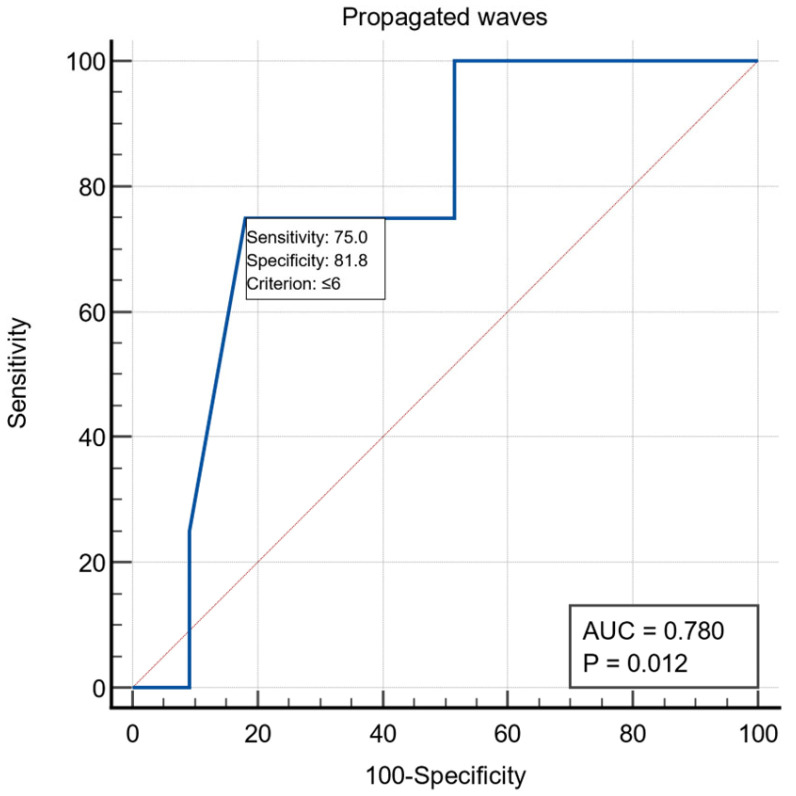
ROC curve for the manometric variable number of propagated waves.

**Figure 2 life-13-00340-f002:**
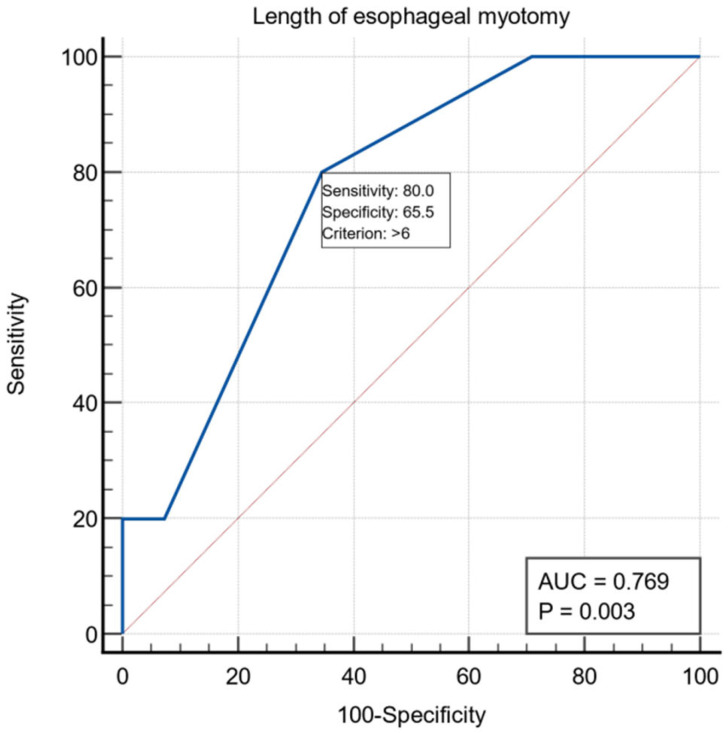
ROC curve for the intraoperative variable length of esophageal myotomy.

**Figure 3 life-13-00340-f003:**
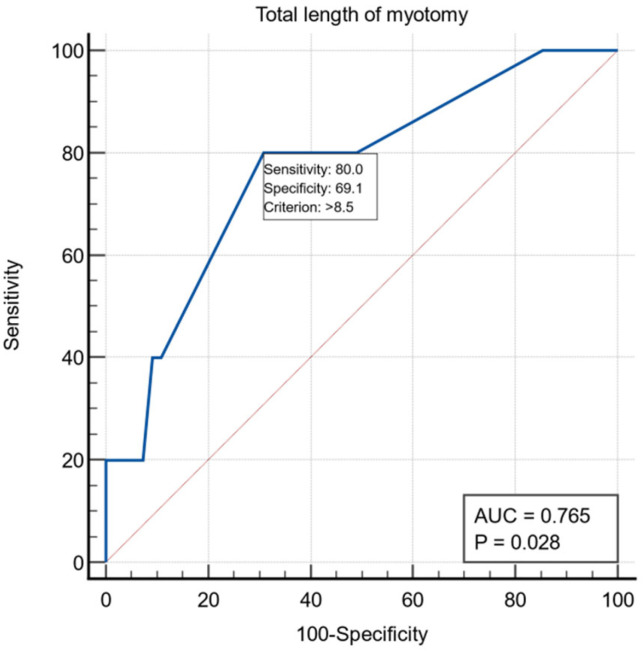
ROC curve for the intraoperative variable total length of myotomy.

**Table 1 life-13-00340-t001:** Demographic, clinical, paraclinical and intraoperative characteristics of patients correlated with intraoperative mucosal perforations.

Intraoperative Mucosal Perforation	No. (*N* = 55)	Yes (*N* = 5)	*p*-Value (Test)
F M	23/55 (41.8%) 32/55 (58.2%)	1/5 (20%) 4/5 (80%)	0.63918 (^1^)
Age	51.2 ± 15.9	41.4 ± 10.6	0.18144 (^2^)
History of abdominal surgery = yes	3/55 (5.5%)	0/5 (0%)	1.00000 (^1^)
Primary DP Yes No	9/55 (16.4%) 46/55 (83.6%)	0/5 (0%) 5/5 (100%)	1.00000 (^1^)
Smoker = yes	18/55 (32.7%)	1/5 (20%)	1.00000 (^1^)
Retrosternal chest pain = yes	18/55 (32.7%)	2/5 (40%)	1.00000 (^1^)
Regurgitation = yes	38/55 (69.1%)	4/5 (80%)	1.00000 (^1^)
Weight loss	26/55 (47.3%)	3/5 (60%)	0.66585 (^1^)
Eckardt score	7.0 [5.0, 8.0]	7.0 [4.5, 9.5]	0.78647 (^3^)
Other symptoms = yes	4/55 (7.3%)	0/5 (0%)	1.00000 (^1^)
Duration of symptoms	21.0 [9.0, 36.0]	12.0 [4.5, 132.]	0.60041 (^3^)
TBE			
Esophageal stasis = yes	34/55 (61.8%)	4/5 (80%)	0.64319 (^1^)
EGJ stenosis = yes	34/55 (61.8%)	4/5 (80%)	0.64319 (^1^)
Esophageal dilatation = yes	31/34 (91.2%)	3/4 (75%)	0.37173 (^1^)
Dilatation grade III No Yes	52/55 (94.5%) 3/55 (5.5%)	4/5 (80%) 1/5 (20%)	0.30059 (^1^)
Tertiary contractions = yes	6/34 (17.6%)	3/4 (75%)	0.03471(^1^)
Barium column (1 min) = yes	33/34 (97.1%)	4/4 (100%)	1.00000 (^1^)
Barium column (5 min) = yes	32/34 (94.1%)	4/4 (100%)	1.00000 (^1^)
UE			
Esophageal stasis = yes	17/55 (30.9%)	2/5 (40%)	0.64769 (^1^)
Esophagitis = yes	5/55 (9.1%)	0/5 (0%)	1.00000 (^1^)
Gastritis = yes	9/55 (16.4%)	0/5 (0%)	1.00000 (^1^)
Manometry			
LES length	4.74 ± 1.71	4.13 ± 0.48	0.48315 (^2^)
LES basal pressure	20.8 ± 8.54	18.0 ± 5.35	0.52168 (^2^)
LES resting baseline	46.5 [32.5, 65.0]	31.0 [27.0, 68.7]	0.29017 (^3^)
LES relaxation (%)	81.0 [47.0, 98.5]	55.0 [47.0, 69.7]	0.22826 (^3^)
Relaxation period	1.00 [0.50, 3.40]	0.60 [0.45, 1.05]	0.24957(^3^)
Wave amplitude > 35 mmHg	56.21 ± 26.905	57.3 ± 29.2	0.94276 (^2^)
Propagated waves ≤6	6/33 (18.2%)	3/4 (75%)	0.03752(^1^)
Vector Volume	6055 [2352.5, 10077.5]	3239.5 [2353.5, 8183.0]	0.52487 (^3^)
Type of achalasia 1 2	15/44 (34.1%) 29/44 (65.9%)	1/4 (25%) 3/4 (75%)	1.00000 (^1^)
ASA risk 1 2 3 4	7/55 (12.7%) 27/55 (49.1%) 19/55 (34.5%) 2/55 (3.6%)	2/5 (40%) 2/5 (40%) 1/5 (20%) 0/5 (0%)	0.49557 (^4^)
Intraoperative data			
Length of esophageal myotomy	6.0 [5.0, 7.0]	7.0 [6.5, 10.0]	0.03961 (^3^)
Length of gastric myotomy	2.0 [2.0, 3.0]	2.0 [2.0, 3.0]	0.63586 (^3^)
The total length of myotomy	8.0 [8.0, 9.0]	9.0 [8.5, 13.0]	0.04423 (^3^)
AR procedure Dorr Toupet	54/55 (98.2%) 1/55 (1.8%)	4/5 (80%) 1/5 (20%)	0.16102 (^1^)
Associated surgery = yes	6/55 (10.9%)	1/5 (20%)	0.47456 (^1^)
Methylene blue = Yes	3/55 (5.5%)	4/5 (80%)	0.00034 (^1^)
Intraoperative endoscopy = yes	47/54 (87%)	1/5 (20%)	0.00326 (^1^)
Intraoperative manometry = yes	1/55 (1.8%)	1/5 (20%)	0.16102(^1^)
Place of perforation Esophagus Cardia	-	4/5 (80.0%) 1/5 (20.0%)	-
Postoperative data			
Days in ICU	1.0 [1.0, 1.0]	1.0 [1.0, 1.0]	0.31826 (^3^)
Complications = yes	1/55 (1.8%)	1/5 (20%)	0.16102 (^1^)
Reintervention = yes	1/55 (1.8%)	1/5 (20%)	0.16102 (^1^)
Postoperative days of hospitalization	3.5 [2.25, 4.75]	8.0 [5.5, 9.0]	0.00137 (^3^)
Days of hospitalization	6.0 [4.0, 9.0]	10.0 [7.0, 12.5]	0.03437 (^3^)
Follow up data			
Postoperative Eckardt Score	2.0 [1.0, 2.0]	0.0 [0.0, 1.5]	0.07936 (^3^)
Δ Eckardt Score	5.09 ± 2.12	6.40 ± 2.79	0.20216 (^2^)

^1^ Fisher’s Exact Test; ^2^ Independent Samples T Test; ^3^ Mann–Whitney U; ^4^ Likelihood Ratio; PD—pneumatic dilatation; TBE—timed barium esophagogram; EGJ—esophagogastric junction; UE—upper endoscopy; ASA risk—American Society of Anesthesiologists risk; AR procedure—anti-reflux procedure; ICU—Intensive Care Unit; Δ EckardtScore—the difference between pre and postoperative Eckardt score.

**Table 2 life-13-00340-t002:** Univariate statistical regression analysis of intraoperative mucosal perforations.

	Univariate Logistic Regression	
	OR (95%CI)	*p*-Value
Tertiary contractions = yes	14.0 (1.23, 159)	0.03321
Propagated waves ≤ 6	14.5 (1.18, 153)	0.03579
Length of esophageal myotomy	1.74 (1.04, 2.89)	0.03196
The total length of myotomy	1.74 (1.04, 2.94)	0.03478
Intraoperative endoscopy = yes	0.04 (0.01, 0.38)	0.00566

## Data Availability

Data supporting reported results can be found on request at correspondent authors.

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
