# Peer review of "Laparoscopic Esocardiomyotomy—Risk Factors and Implications of Intraoperative Mucosal Perforation"

_life, 2023, doi:10.3390/life13020340_

Round 1
Reviewer 1 Report
1. Reference no 1. need correction - Author numbers as Per journal requirement.
2. In Univariate statistical regression analysis factors like tertiary contraction , propagated waves has wide dispersion of CI ( less certain ) even though P Values are significant - this need to be considered while concluding the study findings.
Author Response
Reviewer 1
Reviewer’s comment
Re1. Reference no 1. need correction - Author numbers as Per journal requirement.
Response
Thank you for your remark.
We have modified all references, according to the Per journal requirement.
Reviewer’s comment
- In Univariate statistical regression analysis factors like tertiary contraction , propagated waves has wide dispersion of CI ( less certain ) even though P Values are significant - this need to be considered while concluding the study findings.
Response
Thank you for your comment.
The 2 variables (tertiary contraction=yes, propagated waves≤6) were present in 75%, respectivelly of the patients who presented the observed event (mucosal perforation), therefore an OR of apox 14 was obtained which explains the wide confidence interval obtained. The P_value of close to the limit of statistical significance is explained by the fact that the number of events that occurred was small (5). We have emphasized this aspect in the limitations of the study by the next addition marked in red.
The limitations of the study may be related to the mono-institutional, retrospective nature of a relatively small sample size, with a small number of events (mucosal perforations).
Reviewer 2 Report
The work by Alkadour A. et al is a retrospective cohort study to determine: 1. risk factors of mucosal perforation during laparoscopic Heller`s myotomy and 2. its clinical implications. The idea of the study is to identify risk factors to decrease their incidence and make this surgery safer. The study is well explained, and the findings are in line with the current literature, but I had a few comments: - 1. In the Abstract section in the Material and Methods -Authors should mention what kind of collected data were analyzed (i.e. preoaperative scores, manometric studies, TBE... ) 2. Introduction section is too long- Section about baloon dilatation complications is not necessary 3. In the analyzed group of patients there was surprisingly low rate of preoperative baloon dilatations - only 9 patients, according to literature a individualizad therapy conducted by interventional gastroenterologist can help optimize targeted treatment of achalasia subtypes. Of course this is centre- specific but if the Authors could clarify the multidisciplinary approach to those patients - it would give additional value and resolve suspition of not including of more severely diseased patients. Also section of the centre specific postoperative ambulatory care follow-up in the term of recurrence of symptoms screening should be a little more clarified. 4. Authors are from the Surgical centre so in my opinion, despite standarisation of procedure, more centre specific surgical details might be added- whether or not it was laparoscopy 2D or 3D? Centre specific preoperative patients preparation- it my potentially have impact on generally good outcomes. 5. In the Discussion Section authors could add just in short comment a potential surgical benefits from new technical improvements like: robotic-assisted cardiomyotomy- underlined by US authors.
Author Response
Reviewer 2
Reviewer’s comment
The work by Alkadour A. et al is a retrospective cohort study to determine: 1. risk factors of mucosal perforation during laparoscopic Heller`s myotomy and 2. its clinical implications. The idea of the study is to identify risk factors to decrease their incidence and make this surgery safer. The study is well explained, and the findings are in line with the current literature, but I had a few comments:
Reviewer’s comment
- 1. In the Abstract section in the Material and Methods -Authors should mention what kind of collected data were analyzed (i.e. preoaperative scores, manometric studies, TBE... )
Response
Thank you for your question.
We have added the following information, marked in red.
Material and methods: We retrospectively identified the patients with laparoscopic esocardiomyotomy performed at Sf. Maria Hospital Bucharest, in the period between January 2017- January 2022 and collected the data (preoperative – clinic, manometric and imaging, intra- and postoperative).
Reviewer’s comment
- Introduction section is too long- Section about baloon dilatation complications is not necessary
Response
Thank you for your comment.
We have made the following modifications, marked in red:
During any invasive procedure, there are some risks, the most important being esophageal perforation. After endoscopic balloon dilatation, esophageal perforation is a severe complication, with an overall median rate of 1.9% (range 0%–16%), in experienced hands (>100 patients treated) [5], solving this complication involving various methods [6, 7]
The intraprocedural diagnosis of this complication, allows the prompt establish-ment of various therapeutic options, from conservative treatment, antibiotics plus ob-servation, endoscopic closure with clips or esophageal stent [6], a method limited by the large esophageal lumen of the patient with achalasia, which facilitates stent migration, to surgical intervention, abdominal or thoracic, open or minimally invasive [7].
Reviewer’s comment
- In the analyzed group of patients there was surprisingly low rate of preoperative baloon dilatations - only 9 patients, according to literature a individualizad therapy conducted by interventional gastroenterologist can help optimize targeted treatment of achalasia subtypes. Of course this is centre- specific but if the Authors could clarify the multidisciplinary approach to those patients - it would give additional value and resolve suspition of not including of more severely diseased patients.
Response
Thank you for your comment. We have added the following paragraph amarked in red:
The decision of surgical intervention was established in the multidisciplinary team, the young patients (<40 years) being proposed directly to the operation, in the case of the older patients it was their option, taking into account the frequent need for several dilation sessions. 10 patients with relapsed achalasia were excluded, being re-operated by open surgery.
Reviewer’s comment
Also section of the centre specific postoperative ambulatory care follow-up in the term of recurrence of symptoms screening should be a little more clarified.
Response
Thank you for your remark.
The patients included in the batch were either readmitted to the clinic for evaluation or contacted by phone, being questioned about the evolution of the symptoms that make up the Eckardt score, 3 months after the surgical intervention.
Reviewer’s comment
- Authors are from the Surgical centre so in my opinion, despite standarisation of procedure, more centre specific surgical details might be added- whether or not it was laparoscopy 2D or 3D?
Response
Thank you for your question. We have added in text that In all the cases we have used 3D laparoscopy, and marked it in red.
Reviewer’s comment
Centre specific preoperative patients preparation- it my potentially have impact on generally good outcomes.
Response
Thank you for your comment.
We have added the next paragraph, marked in red.
In order to empty the esophagus of solid food, we administrated 500 ml carbonated drink at lunch and strictly liquid diet in the day before the operation. Anticoagulant prophylaxis was done using low molecular weight heparine in the evening and the naso-esophageal tube was inserted in the morning of the operation.
Reviewer’s comment
- In the Discussion Section authors could add just in short comment a potential surgical benefits from new technical improvements like: robotic-assisted cardiomyotomy- underlined by US authors.
Response
Thank you for your question.
We have added the following paragraph, marked in red:
The authors of two recent meta-analyses comparing results after laparoscopic and robotic Heller myotomy stated that the only statistically significant difference was intraoperative oesophageal perforation rate.[19, 20]
Reviewer 3 Report
Achalasia is a rare disease with significant symptoms and often requires surgical therapy. The laparoscopic Heller myotomy is a standard procedure but it requires a certain expertise and mucosal perforation is quite often. This can sometimes lead to devastating complications and maximal care has to be taken in order to avoid this complication. Because of this, the current study is on a topic of relevance and general interest to surgeons active in Upper-GI surgery.
In this current retrospective study the authors investigate the possible risk factors for intraoperative mucosal perforation and its implications on the postoperative outcomes in a study collective of 60 patients. The authors reveal several risk factors like the presence of tertiary contractions, the number of propagated waves ≤6, the length of esopha-
geal myotomy, the length of esocardiomyotomy and a protective factor – the intraoperative upper endoscopy. These are interesting findings but need to be verified in prospective studies.
I recommend a minor revision:
Minor comments
1. Section Introduction: Lines 46-57 : please add citations 2. Section Results, Line 203-204: please revise the sentence. The statement about the persistence or recurrence of dysphagia is incomplete /unclear 3. Section Results, Line 206: dysphagia 60/in 60 patients. Please delete „in“ 4. Section Results, Descriptive analysis of the group: please add percentages
Author Response
Achalasia is a rare disease with significant symptoms and often requires surgical therapy. The laparoscopic Heller myotomy is a standard procedure but it requires a certain expertise and mucosal perforation is quite often. This can sometimes lead to devastating complications and maximal care has to be taken in order to avoid this complication. Because of this, the current study is on a topic of relevance and general interest to surgeons active in Upper-GI surgery.
In this current retrospective study the authors investigate the possible risk factors for intraoperative mucosal perforation and its implications on the postoperative outcomes in a study collective of 60 patients. The authors reveal several risk factors like the presence of tertiary contractions, the number of propagated waves ≤6, the length of esophageal myotomy, the length of esocardiomyotomy and a protective factor – the intraoperative upper endoscopy. These are interesting findings but need to be verified in prospective studies.
I recommend a minor revision:
Minor comments
Reviewer’s comment
- Section Introduction: Lines 46-57: please add citations
Response
Thank you for your comment. We have added citations.
Reviewer’s comment
- Section Results, Line 203-204: please revise the sentence. The statement about the persistence or recurrence of dysphagia is incomplete /unclear
Response
Thank you for your comment. We have reformulated the sentence and marked it in red
„and they were sent to surgical treatment due to the persistence or recurrence of dysphagia”
Reviewer’s comment
- Section Results, Line 206: dysphagia 60/in 60 patients. Please delete „in“
Response
Thank you for your comment.
We’ve made the correction of the text and marked in red.
Reviewer’s comment
- Section Results,Descriptive analysis of the group: please add percentages
Response
Thank you for your question.
We’ve added percentages in the text and marked in red.